# Tissue Engineering for Gastrointestinal and Genitourinary Tracts

**DOI:** 10.3390/ijms24010009

**Published:** 2022-12-20

**Authors:** Elissa Elia, David Brownell, Stéphane Chabaud, Stéphane Bolduc

**Affiliations:** 1Centre de Recherche en Organogénèse Expérimentale/LOEX, Regenerative Medicine Division, CHU de Québec-Université Laval Research Center, Québec, QC G1J 1Z4, Canada; 2Department of Surgery, Faculty of Medicine, Université Laval, Québec, QC G1V 0A6, Canada

**Keywords:** tissue engineering, gastrointestinal tract, urinary tract, female genital tract, hollow structure

## Abstract

The gastrointestinal and genitourinary tracts share several similarities. Primarily, these tissues are composed of hollow structures lined by an epithelium through which materials need to flow with the help of peristalsis brought by muscle contraction. In the case of the gastrointestinal tract, solid or liquid food must circulate to be digested and absorbed and the waste products eliminated. In the case of the urinary tract, the urine produced by the kidneys must flow to the bladder, where it is stored until its elimination from the body. Finally, in the case of the vagina, it must allow the evacuation of blood during menstruation, accommodate the male sexual organ during coitus, and is the natural way to birth a child. The present review describes the anatomy, pathologies, and treatments of such organs, emphasizing tissue engineering strategies.

## 1. Introduction

The human body can be seen as complex machinery aimed mainly at two primary objectives: first, to keep itself alive and, therefore, to breathe, eat, and drink to obtain nutrients, which will be metabolized to allow the growth and maintenance of tissues. The second function is reproduction. Other human body functions are also essential and effective in a social environment and have made our species successful. Primary functions are present in one form or another in most species with some degree of development. So, from insects to humans, most of these functions will depend on hollow structures. Therefore, the disorganization of these structures can have severe and immediate consequences for humans, and medicine has focused on solving these problems for a long time.

Nevertheless, helpful but non-optimal strategies have often been developed to treat these pathologies. Recently, a new option has appeared which could revolutionize our clinical practices in the future and help relieve the ailments of many patients: tissue engineering. The literature review proposed here will focus on only part of these hollow structures: the digestive and genitourinary tracts. For example, excellent reviews also exist on respiratory tract or blood circulation tissue engineering.

## 2. Tissue Engineering

Tissue engineering is a set of techniques intended to produce tissues or organs in vitro to compensate for the lack of tissues that can be used for transplants, but also to provide more efficient study models for fundamental research. This approach was conceptualized by Langer and Vacanti in a paper that is generally considered the birth certificate of tissue engineering [1]. Synthetic or natural biomaterials, including decellularized organs, may be used as scaffolds and are often seeded with cells (Figure 1). Many techniques can be applied to create the scaffold, such as casting [2], electrospinning [3], or bioprinting [4]. In addition, the self-assembly technique is a reconstruction method which does not require any exogenous biomaterials [5]. 

### 2.1. Materials

Tissue engineering mainly relies on two sets of strategies: scaffold-based and scaffold-free ones (e.g., cell microsphere, cell sheet, or self-assembly) [6]. Scaffold-based TE uses a wide range of materials from silicone rubber [7] to plant leaves [8], including different types of plastics or biocompatible polymers, elements of the extracellular matrix, or even collagen derived from Tilapia skin [9]. Biomaterials need to be biocompatible, have good mechanical properties and be adequate to support cell differentiation. The most challenging steps for biomaterials are to control their degradation rate if the biomaterials have to be resorbed or not, and the sterilization steps can profoundly modify the product’s characteristics [10,11]. A significant interest in biomaterials is to control (or try to control) various parameters, including topological and biochemical parameters, to favour cell adhesion, proliferation and, eventually, differentiation. Biomaterials allow for recreating a chosen form at a reasonable cost and with high reproducibility. Progress has been made to functionalize biomaterials [12]. In the case of decellularized tissues, the extent of preservation of the organ’s extracellular matrix is the key to success [13,14]. The self-assembly technique will be described in more detail below in Section 2.4.

#### 2.1.1. Synthetic Materials

Biomaterials are generally inexpensive and easy to use for engineering but are more prone than natural ones to induce inadequate biological signalling, which allows low or erroneous differentiation of the cells, especially epithelial cells. Moreover, the degradation products of these molecules are generally not recognized by the body and can induce graft rejection or damage [15]. Additionally, lactate, a by-product of many synthetic biopolymers, is admitted to playing a role in cancer progression [16]. Among the available synthetic biomaterials [6,17,18,19,20], the most common are polylactic acid (PLA), polyglycolic acid (PGA), poly(lactic-co-glycolic acid) (PLGA), poly-l-lactic acid (PLLA), polycaprolactone (PCL), poly(hydroxyl butyrate) (PHB), poly(hydroxyalkyl (meth)acrylate) (PHAA), polyglycerol-sebacate (PGS), polydioxanone (PDO), polyethylene, glycol (PEG), polyurethane (PU), polyethylene terephthalate (PET/Dacron), polytetrafluoroethylene (PTFE/Teflon), silicone, vicryl, and nylon. Some biomaterials can be functionalized, especially by grafting peptides, to improve biocompatibility. These materials can also be combined to control parameters such as porosity and degradation rate. Other materials exist, such as smart materials [20,21] or conductive materials [22] with novel features, which can be required for specific applications, e.g., cardiac stimulation.

#### 2.1.2. Natural Materials

Natural biomaterials present in the extracellular matrix (ECM) and used in tissue engineering offer the advantage of being recognized by the cells with no or few modifications. Nevertheless, these biomaterials alone do not necessarily have an organization similar to that of the extracellular matrix and only constitute a part of it, which can induce erroneous signals for the cells. Moreover, these biomaterials rarely possess mechanical properties sufficient for clinical use. Nevertheless, we can combine them to mimic the ECM better and increase their mechanical features. Among them (but not exhaustively) [6,17,19,20] are hyaluronic acid and other glycosaminoglycans (GAG) such as chondroitin–sulfate, collagen, elastin, fibrin, and hydroxyapatite. Other biomaterials found in nature can be used as well. These compounds are not found in the mammalian ECM but are generally well tolerated and degrade without producing a reaction in the body. For example, this category can be found in rubber, chitosan, agar, alginate, xanthan, starch, cellulose, dextran, and silk.

#### 2.1.3. Hybrid Materials

Synthetic and natural biomaterials all have advantages and disadvantages. Additionally, mixing them can create hybrid materials combining the benefits of both kinds of products. Additionally, we can mix several products of each category (or from only one) to increase the potential of the biomaterials.

#### 2.1.4. Acellular Matrix

Removing cellular components from a matrix can provide the best scaffold possible for the cells’ adhesion, proliferation, and differentiation. All the features of the ECM are the same among humans and very close among animals. However, this promise can only be kept if the decellularization process preserves its composition and architecture, allowing it to play the role of support and be a reservoir of growth factors released with natural kinetics. Many efforts have therefore been undertaken to develop chemical, physical, and/or enzymatic treatments, which will make it possible to eliminate the potentially immunogenic elements while preserving the favourable properties of the matrices [13]. 

### 2.2. Cells

Multiple cell types have been used for tissue reconstruction to produce grafts or study models [23]. We will detail them in the following lines.

#### 2.2.1. Embryonic Stem Cells

Embryonic stem cells, provided they have been extracted and maintained under favourable conditions, should be the most interesting for tissue reconstruction since they have an infinite capacity for division and the possibility of differentiating in all possible lineages [24]. However, in addition to the ethical controversy related to the use of stem cells from embryos [25,26], the price of these cell cultures and potential host reactions following the transplant (even if embryonic stem cells have low immunogenicity), and a number of technical obstacles stand in the way of their use, particularly with regard to the adequate differentiation of these cells [27]. We must realize that having the capacity to produce all the lineages and guiding these cells in the right direction is a relatively tricky task. Even sorting the cells following the differentiation process is not a guarantee of success. This is limited by the factors chosen by the researchers as markers of cellular differentiation but remains necessarily limited in terms of scientific knowledge and technological capacity. Nevertheless, these cells remain potentially attractive for basic research while bearing in mind the limits mentioned above.

#### 2.2.2. Neonatal Stem Cells

The umbilical cord is an ideal choice among several tissues readily available to extract cells. Indeed, from Wharton’s jelly, MSCs (WJ-MSC) have excellent proliferative potential and a high growth rate. These Neonatal stem cells can better retain their potential for differentiation in multiple lineages with passages than other MSCs. [28,29,30,31]. Part of the ethical controversy also exists for these cells, although in this case, it is nevertheless reduced by the fact that destroying an embryo is unnecessary [32]. The problem is more of a social or societal nature in this case. However, there are still significant costs, mainly related to the extraction, amplification, and storage of these cells. Umbilical cords are harvested just after delivery and are a waste product [33]. The extraction of WJ-MSCs does not incur potential side effects as in the protocol used for collecting adult stem cells from bone marrow or adipose tissue. Additionally, the yield of cell extraction from umbilical cord samples is higher for WJ-MSC than for other sources. These cells can be easily expanded with high glucose cell culture medium supplemented with fetal calf serum. We can obtain very efficient separation of MSCs from the Wharton jelly pieces placed in a dry cell culture surface [34]. Although the problem of adequate cell differentiation may be less acute than with embryonic stem cells, we cannot dismiss it. It must always be borne in mind by researchers who want to reconstruct tissues, particularly in the cases of transplants for patients. Nevertheless, WJ-MSCs are not known to produce teratoma after subsequent transplantation [34]. 

#### 2.2.3. Adult Stem Cells

Adult stem cells and progenitor cells are good candidates for tissue engineering [35]. Indeed, these cells have a reduced differentiation potential and are often limited to the organ from which they are extracted [36]. Their maintenance in culture is generally a less delicate process than that of stem cells with more significant differentiation potential. They nevertheless retain an excellent possibility for proliferation. Regulatory agencies currently approve many such cells that are used in clinical trials. Among the most widely used, one can find the mesenchymal stem cells (MSC) [37], especially from bone marrow (BM-MSC), the adipose-derived stem cells (ASC) [38] but also stem/progenitor cells from the urothelial and vaginal epithelium, or urine-derived stem cells (USCs) [23,39], for example. These cells are generally relatively easy to extract and give quite attractive yields in the case of ASCs. MSCs and ASCs are also known for their immunomodulatory power. They can be beneficial during transplantation to limit immune reactions, including in the case of tissue reconstructed with autologous cells [40]. In addition to their use in terms of reconstruction, the potential of these cells has already been shown for the regeneration/repair of tissues, mainly through their paracrine effects [41]. 

#### 2.2.4. Adult Somatic Cells

Autologous adult somatic cells could, a priori, be considered an adequate solution in terms of safety since they come from the patient themselves. No ethical controversy exists regarding the use of these cells. However, with the reversal of their differentiated state, their potential for division is minimal, and therefore the potentially producible tissue surfaces will be correspondingly reduced. Another problem is that cells from diseased organs may also have deficiencies and potentially not be used to rebuild tissue [42]. The use of cells from other organs is always possible, but it is then necessary to wonder about the capacities of these cells to recapitulate the function of the organ that one wants to reconstruct. For example, the use of skin cells could be interesting, but the cutaneous and intestinal functions are clearly different and not necessarily compatible. Nevertheless, it is only possible to extract somatic cells with stem cells from the same biopsy. This is, instead, an advantage because these cells, as indicated above, will have a lower capacity for differentiation but, in general, a good capacity for proliferation. Suppose the somatic cells can maintain the architecture and function of the organ until the adult stem cell fraction of the tissue can take over, in that case, this may be sufficient to obtain a functioning tissue.

#### 2.2.5. Induced Pluripotent Stem Cells

The concept of pluripotent stem cells was introduced in 2006 by the work of Dr. Yamanaka’s team [43]. This work constitutes a real historic breakthrough for scientific research in biology. Nevertheless, as is often the case, the enthusiasm for these works went beyond what they immediately allowed. It is clear that many technical obstacles still exist today and drastically limit their use in a clinical context. The theory around these cells is that the expression of transcription factors by somatic cells can induce the reprogramming of cells into cells with a potential similar to that of embryonic stem cells. The primary interest of these cells is firstly at the ethical level since adult cells from the patient are used, but also, if necessary, the possibility of correcting the genome of these cells before their amplification, giving a new impetus to gene therapy. As with embryonic stem cells, one can nevertheless question the capacity that biologists have to differentiate adequately and with sufficient control of these cells before using them. Second, it appears their differentiation range depends on many factors and is not as “magical” as expected [44,45]. This is without considering the technical difficulty of reproducing the results obtained. The equally critical point is that the somatic cell, which is brought back to its state of pluripotency, has a past of epigenetic or protein modifications, which are not necessarily so easily reversed. We rarely make new out of old, even in biology [46,47]. Many years of study will still be needed to realize the extraordinary potential of these cells. In particular, it will be necessary to understand better the mechanism of reprogramming somatic cells into stem cells, to better understand the stages of differentiation: only persevering work will give long-term results that will benefit patients.

#### 2.2.6. Cell Culture Condition Improvements

Suitable cell culture conditions are necessary for the reconstruction of tissues by tissue engineering. Indeed, it is essential to find not only conditions where fibroblasts, muscle, or endothelial cells can fulfill their roles—for example, allowing the deposition of extracellular matrix or forming a network of capillaries—but also supporting the adequate differentiation of epithelial cells by allowing preservation of stem cells to ensure long-term graft survival.

Contrarily to the physiologic conditions, i.e., when cells are in the human body, standard cell culture conditions expose cells to higher oxygen levels (20% O_2_) than physiologically relevant (called physioxia). The hyperoxic environment of traditional cell culture induces damage, leading to a stress state in cells. Therefore, optimal oxygen concentration, near physioxia, should be used in cell culture to avoid too high oxygen and its subsequent oxidative damage but not too low to allow a good metabolism [48]. For example, one study demonstrated that culture in hypoxia allowed better preservation of the stem/progenitor character of urothelial cells [49] and another that conditioning a basic cell culture medium with fibroblasts in hypoxia was effective in supporting the formation of a well-developed capillary network [50]. Hypoxia can also be used with benefits to expand MSCs. This condition allows the expansion of MSC without morphologic differentiation and has a reduced content in mitochondria.

Another point which should be taken into account by the researcher is the composition of the cell culture medium and especially the use of fetal bovine serum (FBS). Due to its ill-defined and highly variable composition, FBS limits the reproducibility of results depending on the batches used and poses a safety problem for the production of tissue engineering products. Therefore, there is an urgent need to switch from a serum-based medium to a serum-free medium (SFM). Several SFM have been tested and successfully achieved the culture of MSCs [51]. Recently, SFM was tested with positive results for expansion and stem cell preservation for keratinocytes, urothelial cells, and skin and bladder fibroblasts [52]. 

### 2.3. Techniques

Casting is used to produce a variety of geometries and shapes with homogenous cell populations. It consists of pouring or injecting a liquid into a mould to form a solid structure. Solidification of the material is done by gelation, crosslink, or evaporation/lyophilization. Cells can be cast with biomolecular gels such as collagen and fibrin. However, a secondary seeding step may be necessary to create the stratified cellular structure of native tubular tissues. Complex shapes can be cast using various mandrel sizes to obtain different layers and generate proper, tissue-like structures. This method is commonly used due to its compatibility with a wide range of materials and can be applied to produce complex models with stratified layers of cells [2,53]. 

Electrospinning consists of the solubilization of a polymeric or biomolecular material ejected from a syringe. The fluid is charged through an applied voltage and directed toward a neutral or oppositely charged mandrel. The solvent evaporates as the material travels toward the mandrel, creating nanofibers. The mandrel is rotated during the extrusion process in order to form tubular scaffolds. The engineer can control various parameters such as size, porosity, and mechanical properties. Electrospinning is compatible with multiple materials (synthetic and natural biomaterials) suitable for scaffold production [2,53]. 

Another technique consists of the rolling of a flat substrate into a tube. It is accomplished using a mandrel to roll the substrate manually. Tubular structures can be created from either polymeric sheets or ECM sheets obtained by the self-assembly mechanism. In this mechanism, ECM-producing cells are cultured to confluence. Then, the ECM sheet obtained is detached and rolled into a tube, where other cells can be seeded. Once the structure has been rolled, tubes can be closed with stitching or through a sealant to approximate free edges [2]. 

Three-dimensional (3D) printing or bioprinting forms a 3D structure layer-by-layer. Various techniques and adaptations exist for 3D bioprinting, the most prevalent being extrusion-based. This technique is compatible with most material types, depending on the technology used, including hydrogels and hydrogels containing live cellular populations and natural biomaterials, such as collagen and fibrin. Specialized printers may be required depending on the material. The main advantage of 3D printing is the potential for easily customizable scaffolds and the potential for printing materials containing live cells. However, the main challenge is maintaining viable cells in the printed bioink due to heat and mechanical load [2,53]. Bioprinting may be applied to produce ureters and urethra. It allows the high-resolution control of the substitute’s microarchitecture. As a result, tubular constructs with mechanical properties similar to native tissues can be obtained. However, it is still difficult to produce complex substitutes due to their micro- and macro-architectures. The main disadvantages of this technique are the small size of printable scaffolds, the limited bio-inks available, their cost, and the long processing time [53]. 

Decellularization: Decellularization consists of the production of a scaffold from native tissue. The tissue is treated with a combination of enzymes, detergents, buffers, and freeze–thaw cycles to lyse and remove cells. Decellularized scaffolds may be reseeded with an appropriate cellular population and used for allograft/xenograft transplantation or implantation. Decellularization has many advantages, mainly biocompatibility and mechanical properties [2]. Although decellularized scaffolds appear to have more benefits than others, they represent some disadvantages. Residues of biological elements in these tissues, such as DNA or prion, may cause an immune risk which could be avoided by more specific decellularization protocols [53]. 

### 2.4. The Self-Assembly Technique

Preformed scaffold requires the use of biomaterials, synthetic, hybrid, or natural. It could be a disadvantage if it represents an obstacle to adequate differentiation of the cells or induces an unwanted body reaction. Nevertheless, engineered human-derived tissue without exogenous biomaterials to form scaffolds is possible. Such a technique derived from the discovery that ascorbate is a cofactor of the enzyme propyl-4-hydroxylase to produce and assemble collagen fibres and thus is necessary to form the extracellular matrix (ECM) [54]. In the early 1970s, it was demonstrated that human skin fibroblast culture allowed collagen synthesis in the presence of ascorbate (vitamin C) [55]. Seventeen years later, a study showed that adding ascorbate to skin fibroblast culture allowed collagen accumulation to form a three-dimensional tissue-like substance [56]. A tissue engineering technique was then developed, in the mid-1990s by the group of Dr. François A. Auger and Lucie Germain to produce a totally human blood vessel [57]. Following this discovery, bilayered skin substitutes for severely burned patients were reconstructed. Due to the composition and organization of the ECM close to the native tissue and using organ-specific cells, a high level of epithelial differentiation can be achieved [58]. Several other tissues were developed during the following two decades, such as the cornea, fatty tissues, urological tissues, vagina, neural tubes, or cardiac valves [5]. 

Fundamental study models were also developed from the tissues produced by the self-assembly technique to understand pathologies in a context close to native human tissue. Indeed, current models used to study complex diseases, such as cancer or psoriasis, are performed using 2D cell cultures or animals, which inadequately mimic the physiology and are associated with a low potential for clinical translation [59]. Two-dimensional cell cultures on plastic petri dishes generally do not recreate the complexity of a 3D environment, especially cell–cell and cell–ECM interactions. Animal models are often too complex and different from humans [60]. Nevertheless, the contribution to advancing the science of 2D cell cultures and animal models should not be underestimated. More sophisticated models need to be developed to go forward. In this case, tissue engineering, especially the self-assembly method, can be helpful. For example, models have been developed to study skin pathologies such as hypertrophic scars [61], systemic sclerosis (scleroderma) [62], melanoma [63,64], psoriasis [65], epidermolysis bullosa [66], neurofibromatosis [67], or skin manifestations of amyotrophic lateral sclerosis (ALS) [68], but also genitourinary pathologies such as vaginal mucosa infection by the human immunodeficiency virus (HIV) [69], ketamine-induced cystitis [70], bladder cancer [71], or urinary tract infection [72]. 

Protocol: First, fibroblasts and mesenchymal cells are cultured in the presence of ascorbic acid, also known as vitamin C. In the case of bladder reconstruction, it is recommended to use organ-specific mesenchymal cells such as bladder mesenchymal cells. Endothelial cells from the organ-specific microvascular network can be seeded at this initial step to form a capillary-like network. It is also possible to seed immune cells, such as macrophages, to produce immunocompetent models interesting for pre-clinical studies. A reseeding step can be performed two weeks later to improve tissue thickness and cell distribution. Three to six weeks later, a thick sheet results from the extracellular matrix (ECM) deposition by the mesenchymal cells. At this step, epithelial cells such as urothelial cells extracted from a bladder biopsy can be seeded on the top of the construct for one week. Then, the flat construct is raised at the air/liquid interface for three weeks in order to stimulate the differentiation of urothelial cells leading to the formation and maturation of the epithelium. The final three-dimensional tissue can be easily grafted [73]. The flat tissue model can be used in reconstructions that involve only a section of the circumference of tubular organs. In rebuilding whole tubular structures, such as ureters and urethras, the tubular self-assembly technique is adapted (Figure 2). Epithelial and endothelial cells are not seeded directly after the stroma production. The tissue-like sheet is detached from the petri dish and rolled around a mandrel for the formation of a tubular shape. Mesenchymal cells can also be seeded directly on the mandrel to form a tubular structure, avoiding delamination of the rolls in the case of their incomplete fusion. After the formation of the tube, the mandrel is removed. The resulting lumen is perfused with culture media to avoid collapse. Epithelial or endothelial cells can be seeded inside the tube for urethral and ureteral reconstruction. During the epithelial–endothelial cell seeding step, the tube is continually rotated to ensure a uniform distribution of the cells [73]. The main advantage of the self-assembly method is the reduction in inflammatory reactions due to the use of autologous organ-specific cells. However, the high cost and the long time required make this technique difficult to apply [53]. 

## 3. Digestive Tract

### 3.1. Digestive Tract

The anatomy of the digestive tract (Figure 3) is relatively similar throughout. First, an adventitia/subserosa surrounds the other layers. Then muscular layers, with longitudinal and circular orientation, ensure the peristaltic movements that allow the food and the waste products of digestion to progress throughout the digestive tract. Then, the connective tissue on which lays the epithelium can differ notably according to the parts of the system.

#### 3.1.1. Esophagus

The esophagus is the organ which connects the pharynx to the stomach. It is a tubular organ, and its role is to convey nutrients into the stomach. Like the other parts of the digestive tract, it consists of the four layers described above: adventice, muscularis, submucosa, and mucosa. In addition to connective tissue and a vascular network, the submucosa comprises some glands which secrete mucus to help nutriments progress throughout the esophagus via peristalsis. The esophagus is limited by two sphincters, the upper esophagus sphincter and the lower esophagus sphincter. The esophageal epithelium consists of two layers. A basal layer is laid on the basement membrane. It comprises progenitor/stem cells and a suprabasal layer with cells at different differentiation [74]: epithelium and underlying connective tissue form papillae. The niche of progenitor/stem cells is located in the flat interpapillary zone [75]. 

The most common esophageal pathology in the pediatric population is esophageal atresia with or without tracheoesophageal fistula (TEF). The TEF connects the upper part of the esophagus to the trachea, which connects to the lower esophagus. TEF can be proximal or distal, depending on the location of the connections. In some esophageal atresia, the trachea is used instead of the upper part of the esophagus. Esophageal atresia is found in 1/2500–1/5000 births with a variable survival rate but higher in a hospital where neonatal care is developed. Currently, esophageal atresia is treated through a surgical operation consisting of anastomosis of esophageal segments or esophageal to gastric elements and a repair of the esophageal and tracheal tissues to close the fistula. The exact surgical protocol depends on the nature of the defects. Some postoperative complications can happen, such as leaks, stricture, and reflux [76]. In the adult population, the most common pathology is esophageal cancer, a pathology with increasing incidence [77]. The 5-year survival rate is as low as around 20%. The role of several factors, such as genetics and lifestyle, is known, but as it is for most cancers, smoking is a clear risk factor to develop esophagus cancer. Alcohol consumption and gastroesophageal reflux also play a role [78]. 

Treatments depend on the stage of the disease, and endoscopic resection of the esophageal lesion can be done, or chemoradiotherapy can be used as a treatment. Still, esophagectomy remains the treatment of choice for most of these cancers [79]. When the replacement or repair of a sizeable esophageal defect is required, stomach (esophagogastrostomy), jejunum, or colon (esophagocolostomy) tissues can be used [80,81,82]. 

Regenerative medicine can help treat esophageal disease by providing alternative therapy with a better outcome for patients. Tissue engineering can provide different kinds of tissue, more or less complex, to replace or repair damaged esophagus [83]. One challenge of organ reconstruction using tissue engineering is to mimic the niche of stem cells. This case mimics the interpapillary zone’s physical, biochemical, and mechanical features.

#### 3.1.2. Stomach

At the end of the esophagus takes place the gastroesophageal junction which separates the esophagus from the stomach. The stomach’s digestive function is to prepare the food for subsequent digestion steps and nutrient absorption in the intestine, especially by mixing the food in a particular chemical and enzymatic environment. Therefore, the stomach could accommodate the food by flattening the rugae formed mainly in the corpus/fundus. The stomach is divided into three parts: the cardia or cardiac area; the fundus, also named corpus, which is the main part of the stomach, and finally, near the pyloric sphincter separating the duodenum and stomach, the antrum could be found [84]. The histology of the stomach resembles the one of the esophagus with four distinct layers: the mucosa, the submucosa, the muscularis, and finally, the adventice named the subserosa. The mucosa consists in the epithelium, which forms the gastric pits in which there are three to five gastric glands. Several cells can populate the glands: the parietal, the foveolar, the chief cells, and the neuroendocrine G-cells [84]. The parietal cells secrete hydrochloric acid, which is used in the digestive process. The foveolar cells secrete the mucus, a complex glycoprotein mix, to protect the epithelium from this acidity and lubricate the surface to favour the movement of the food through the stomach. The chief cells secrete the pepsinogen. Finally, the G-cells secrete gastrin, a hormone which partially controls the release of digestive secretion. The epithelial cells are derived from stem cells located in the upper middle of the fundic gland over the parietal cells and chief cells at the base, whereas the stem cells are located in the lower center and part at the base of antral glands [85]. 

The mucus-secreting glands are mainly found in the cardiac area and the pyloric mucosa. The corpus/fundus mucosa is primarily dedicated to digestive secretion such as acid and pepsinogen. Contrary to what is observed for most epithelium junctions where the transition is abrupt, e.g., the gastroesophageal junction, the intersections between the different gastric epithelia are gradual [84]. The difference between the other stomach areas can explain the patchy configuration of many gastric diseases. The epithelium is supported by the lamina propria, a connective tissue. At the base of the lamina propria is a thin layer of muscles, the muscularis mucosae. The submucosa contains the main vascular network and nerves, as in the esophagus. The muscularis is composed of three layers, oblique, longitudinal, and circular.

The primary gastric pathologies are gastric ulceration, which can result from progressive tissue erosion and eventually evolve into perforation, and gastric cancer, causing 800,000 deaths yearly [86]. In addition, many pathologies are due to Helicobacter pylori infection, 75–88% of attributable risk, and tobacco and alcohol consumption, which are also negative factors for esophageal pathologies [86]. Several antimicrobial agents are commonly used to eradicate persistent Helicobacter pylori infection, which can help limit cancer progression [86]. Nevertheless, the diagnosis of gastric cancer is often made too late for infection control to have an important role, especially in developing countries where 80% of the population is infected. Coinfection with Epstein–Barr virus, a widespread virus, can also be a causative agent of gastric cancer (10% of the cases). Surgical resection is the principal treatment combined with perioperative/neoadjuvant or adjuvant chemotherapy. Standard gastrectomy implies the removal of two-thirds of the stomach.

#### 3.1.3. The Intestines

The intestines’ role is to digest and absorb the food after its passage through the stomach. The intestines also have secretory and inflammatory functions. Now, the role of intestines and especially their neuronal network, the enteric nervous system, sometimes called the second brain, is established in many pathologies. The role of the intestinal microbiota is also known as essential for human health. The intestine comprises two major segments: the small intestine and the colon. Each of them can be subdivided into smaller parts. The small intestine [87] is generally around 5 m long and 3 cm in diameter, but the length/width can vary [88]. The small intestine begins after the pylore and consists of three parts. The first part is the duodenum, a short section (20–25 cm). Connections with the pancreas and liver allow the continuation of the digestive process started in the stomach. The Brunner’s glands secrete an alkaline fluid which helps to control the acidity of the bowel arriving from the stomach [89]. The second section of the small intestine is the jejunum (2 m). Its function is to absorb nutrients, and villi significantly increase its absorbent surface. The last section is the ileum (3 m) and continues the absorption process of the jejunum. It is connected to the colon through the ileocecal junction. The large intestine (from 9 cm in diameter in the caecum to 6 cm in the transverse colon) begins at this junction and can also be divided into several parts [90]. The caecum is the first section. It is, in some way, the antechamber of the large intestine. The caecum is also associated with the appendix, an organ still little known but which seems to play a role in connection with the lymphoid tissue and potentially serve as a microbiota reservoir. The colon (1.6 m) [91] could also be divided into ascending, transverse, descending, and sigmoid colons. Its structure seems segmented into saccules (haustra), and its central role is recycling water and some residual nutrients from the bowel, especially the ones produced by the microbial flora. It also compacts and stores the feces in the bowel. The end of the colon participates in holding the feces between defecations. The rectum (12 cm long) [92] stores the formed feces (the ultimate waste) until their elimination through the anus, which is the end of the gastrointestinal tract.

As the other part of the gastrointestinal tract, the intestine is a multilayer organ: the mucosa, for absorption of nutrients through a capillary network in the lamina propria; the submucosa, which is a connective tissue containing a lymphovascular network supporting the epithelium, the muscularis, inner circular and outer longitudinal, responsible of the peristalsis, and the adventice. The epithelium is different from the other parts due to its functions. Intestines are highly vascularized and innervated tissues. The intestine consists of a multitude of villi and invaginated crypts. In the small intestine, the crypts contain the stem cells in their bases [85]. By migrating from the base of the crypt to the apex of the villus, these cells differentiate into enterocytes or enteroendocrine, goblet (mucus secretory), and Paneth cells, the first being absorptive cells and the three seconds being secretory cells. As in the small intestine, stem cells are in the base of the crypt and migrate to the luminal surface. Adequate maintenance of intestinal stem cell niches is essential due to the rapid turnover of the intestinal epithelium. Renewal of the intestinal epithelial layer occurs every 4–5 days [93]. The organization of the basement membrane of the intestinal epithelium also seems essential for differentiating the cells into the various cell types required for efficient intestines [94]. 

Many pathologies can affect the intestines, but tissue engineering cannot resolve them all. Short bowel syndrome (SBS) is the most apparent indication affecting the small intestine [95]. Tissue engineering can also be a therapeutic option for diseases which need the removal of a substantial part of the intestines, such as inflammatory bowel diseases [96], including Crohn’s disease and ulcerative colitis, but also cancers. SBS corresponds to an inadequate functional intestinal epithelium or its drastic reduction in surface following surgical resection. The survival rate is as low as 20% at five years in the most severe cases where 10% of the intestine’s average length is present [97]. The transplantation of the small intestinal segment can increase the survival rate to 58% at five years [98], a survival rate which remains unacceptable for children. Nevertheless, the shortage of available tissue to repair the small intestine is also a problem, and this kind of therapeutic solution could not be applied to all patients. By providing a virtually large number of tissues, tissue engineering could meet the needs of clinicians and patients.

#### 3.1.4. Tissue Engineering

The reconstruction of the esophagus (Table 1, Table 2 and Table 3) is less complex than that of an entire stomach or a long segment of the intestine. Indeed, the esophageal epithelium is relatively uniform. For example, the repair of superficial esophageal lesions by the cell sheet technology has been tested as reviewed in [99]. However, there remains the problem of producing a strong enough muscle and, above all, whose contractions can be coordinated to ensure peristalsis. A long segment of the esophagus would obviously pose more of a problem than a short segment. However, research in this area has yielded exciting results. The challenge is to move from reconstruction limited to trials to reconstruction for a large number of patients, notably by limiting the costs of the product to remain accessible. Contrarily, the reconstruction of the gastroesophageal junction will undoubtedly pose more problems [100,101]. Indeed, such a reconstruction would imply a particular architecture and the creation of a transition zone between the epithelia.

In addition to Table 1, Table 2 and Table 3, readers may profit from reading several review articles [83,172,173,174,175,176,177,178]. 

Engineering of the stomach (Table 4) poses obvious problems related primarily to the diversity of cell types and their spatial organization in the organ. Additionally, we need to obtain a high degree of differentiation of epithelial cells to effectively protect the body during digestion and a strongly acidic environment. The most fruitful results have come from the reconstruction of stomach patches [177]. Progress in the knowledge of stem cells, the organization of their niches, and the processes of differentiation into various cell types are therefore necessary to reconstruct this tissue. It will also be interesting to see if associated structures in space, perhaps using bio-printing, would make it possible to obtain exciting results on this point.

In addition to Table 4, readers may profit from reading some review articles [177,200]. 

Reconstruction of the intestine using tissue engineering (Table 5) should be more accessible than the stomach. The intestines also have a great diversity of epithelial cells. Still, the structure is tubular and linear, offering a better possibility of succeeding throughout the reconstructed segments of the functional units. Stomach or intestinal organoids make it possible to reproduce crypts or wells satisfactorily, with a recapitulation of the organization of the different epithelial types, which is very reminiscent of what is found in vivo. Nevertheless, it is evident that even more than for the esophagus, it will be necessary to have a coordinated musculature to ensure the most normal intestinal transit possible through the zones reconstructed by tissue engineering [201]. This coordination will also involve a network to control, which can be either artificial or reconstructed. The importance of the enteric nervous system in many biological mechanisms should encourage research to integrate this network into reconstructed tissues [202,203,204,205]. Finally, there is exciting work in reconstructing the internal anal sphincter, which makes it possible to restore continence in affected patients [206]. 

In addition to Table 5, readers may profit from reading several review articles [176,177,178,230,231,232,233]. 

Most strategies used to reconstruct stomach patches or intestine segments have used either organoid in systems without scaffolding or biomaterials. Although exciting results have been obtained using the self-assembly technique to reconstruct tubular structures such as blood vessels, urethras, or even vaginal mucous membranes, this technique has not been used for stomach or intestinal reconstruction. However, it could prove interesting as it has demonstrated its potential to produce well-organized epithelia when cells from the organ to be reconstructed are used. Moreover, this technique seems particularly suitable for reconstructing the esophagus, which presents essential similarities to those which have already been rebuilt [57,234,235]. This strategy is also interesting because it has demonstrated a strong potential to be prevascularized and thus allows rapid reconnection of the network of the graft to that of the host [236,237,238]. 

Moreover, given the weakly immunogenic nature of the scaffolds produced with this technique [239], it could be envisaged to create ready-to-use acellularized tubular structures. That would allow rapid and usually efficient recellularization. The high similarity in the composition and organization of extracellular matrices is produced by the self-assembly method, particularly when using organ-specific cells.

The fact that there is a great diversity in tissue engineering strategies could allow, in the long term, the development of non-competing products adapted to different clinical situations. For example, it is not necessary to have a preliminary cellularization to repair the short section, although that can help. This cellularization seems essential when exceeding a certain length (more than 2 cm). Similarly, coordinated peristaltic contractions are not necessary for the esophagus’s functioning after a transplant to replace a short segment. Still, they remain essential as soon as significant reconstructions have to be carried out.

## 4. Urinary Tract

### 4.1. Ureters

The ureters are a pair of muscular tubes that connect the renal pelvis with the bladder allowing the unidirectional movement of urine. Each ureter consists of two tissue compartments: the urothelium, which lines the tubular lumen, is a specialized epithelium consisting of a layer of basal cells, one or several layers of intermediate cells, and a luminal layer of superficial cells. The outer mesenchymal coat, which supports the rigidity and flexibility of the tube, is organized in an inner ring of fibroelastic material, the lamina propria, multiple layers of smooth muscle cells, and an outer ring of connective tissue, the tunica adventitia. The tunica adventitia contains blood and lymph vessels and nerves that serve the smooth muscle layers and the urothelium. Ureters actively convey the urine down to the bladder using the peristaltically active smooth muscle layer that, together with fibroelastic material, ensheathes a water-impermeable multilayered urothelium. Unidirectional peristaltic contractions are triggered by pacemaker cells in the pelvic–kidney junction [240]. 

Urinary tract endometriosis (UTE) is defined by the implantation of the stroma or endometrial glandular epithelium outside the endometrial cavity and the uterine musculature, penetrating the retroperitoneal space or the wall of the pelvic organs. The incidence of UTE ranges from 0.3 to 12% of all women affected by endometriosis. The ureter is the second most common site affected by UTE. Ureteral endometriosis (UE) occurs in 0.1–1% of cases. UE is an asymptomatic disease which might lead to the silent loss of renal function. However, nonspecific symptoms may appear, such as flank or abdominal pain, hematuria, renal colic, and hypertension. UE most commonly affects the distal segment of the ureter, less commonly the mid-ureter, and, rarely, the proximal ureter. UE is usually unilateral, but bilateral involvement occurs in 10–20% of cases [241]. Multifocal fibrosclerosis is a rare syndrome characterized by fibrosis involving multiple organ systems. The ureters are the most commonly affected organs, and the disease is sometimes called periureteral fibrosis. Acute anuria and chronic obstruction with azotemia, gross hematuria, and hypertension secondary to obstruction of renal vessels are frequent manifestations of the disease [242]. 

Primary small cell carcinoma of the urinary tract is a rare cancer, primarily localized in the bladder and prostate. Its localization in the ureter is extremely rare but aggressive. The main risk factors are smoking/smoke exposure, age, male sex, tumor size, nodes, metastasis grading, and size at diagnosis [243]. Primary tumors of the ureter account for only 6% of all tumors of the upper urinary tract. Hematuria, frequency, dysuria, and pain are the most frequent symptoms. The leading histologic type is transitional cell carcinoma, which most commonly metastasizes to the regional lymph nodes, with a higher prevalence in individuals who smoke tobacco and those exposed to chemicals used in the chemical dye, rubber, gas, and plastic industries. Secondary ureteral neoplasms occur either by direct extension from an adjacent primary tumor, such as lymphoma, from an adjacent metastatic deposit, or by metastasis from a distant primary tumor. Direct metastasis to the ureter occurs via the blood or lymphatic vessels. The most common sites of primary tumors metastasizing to the ureter include the breast, gastrointestinal tract, prostate, cervix, and kidney. The less common primary sites include the lung, skin, uterus, ovary, and testis [244]. 

A ureteral hernia is uncommon and nearly always indirect, with two types: intraperitoneal and extraperitoneal. It can include the ureter alone or, frequently, other abdominal sliding organs within the hernia sac, such as the bladder and bowel tracts. The most common urinary symptoms are dysuria, frequency, and urgency [245]. 

### 4.2. Bladder

The bladder is an extraperitoneal muscular organ serving as a reservoir that collects and stores urine at low intravesical pressures. It receives urine from the kidneys via the ureters and expels it along the urethra [246]. The bladder wall consists of the bladder mucosa, the detrusor (muscular layer), and the adventice (fat tissue). The bladder mucosa consists of the urothelium, the basement membrane, and the lamina propria (LP), also referred to as the suburothelium. The LP, laying between the basement membrane and the detrusor muscle, is composed of an extracellular matrix containing different cell types, such as fibroblasts, adipocytes, interstitial cells, and afferent and efferent nerve endings. The LP contains blood and lymph vessels, elastic fibres, and smooth muscle bundles (muscularis mucosae) [247]. The LP acts as the capacitance layer of the bladder wall. It determines bladder compliance, which is the ability to fill with urine at low intravesical pressure [246]. It also enables adaptive changes to increasing volumes, while the detrusor functions as the limiting layer to prevent overdistension. In addition, the LP has a role in signal transduction to the central nervous system (nociception, mechanosensation) and may be a source for the production of factors influencing the growth of the urothelium and the detrusor muscle [247]. 

The most common bladder pathologies are bladder exstrophy, neurogenic bladder, bladder pain syndrome, cystitis, and bladder cancer. Bladder exstrophy is a rare congenital malformation in which the bladder is exposed outside the body, with an overall incidence estimated to be between 1 in 10,000 and 1 in 50,000 live births [248]. A neurogenic bladder is caused by neurologic disorders leading to complications, including urinary incontinence [249]. Bladder pain syndrome, also known as interstitial cystitis, may be due to a decreased glycosaminoglycan (GAG) layer, altered urothelium permeability, uroinflammation, or neural upregulation. Its main symptoms are a high frequency of urination, pain localized to the bladder, pelvis, and abdomen, and pain during urination. Interstitial cystitis occurs in 3–8 million women and 1–4 million men [250]. Cystitis is a bacterial infection of the bladder, developed due to the colonization of the periurethral mucosa by bacteria, such as Escherichia coli, from the fecal or vaginal flora and the ascension of such pathogens to the bladder [251]. Urinary tract infections are four times more frequent in females than males, occurring between the ages of 16 and 35 years, with 10% of women getting an infection yearly and more than 40% to 60% having an infection at least once in their lives [252]. 

Bladder cancer is one of the most common cancers worldwide, with an incidence four times higher in men than in women. However, cancers diagnosed in women are more invasive. In addition, advanced age and cigarette smoking may contribute to the development of bladder cancer [253]. The main symptoms are frequent urination with pain, blood in the urine, and lower back pain. Urothelial carcinoma is the type of bladder cancer with the highest risk of death, accounting for 90% of diagnosed cases [254]. 

### 4.3. Urethra

The urethra connects the bladder to the external environment and allows urine excretion from the body. In addition, the male urethra provides a conduit for ejaculation to pass from the distal portions of the male reproductive system, specifically the vas deferens, seminal vesicles, and prostate [255]. 

The histological structure of the urethra consists of three different layers: the epithelium, the submucosal layer, and the fibromuscular layer. The lumen of the male urethra is lined by various epithelia that are protective against constant exposure to urine, seminal fluid, and the external environment. The urothelium covers the prostatic segment, and a stratified or pseudostratified epithelium covers the membranous and penile urethra [256]. 

The submucosal layer is supportive due to its extensive vascular content. The fibromuscular layer is the outer layer that provides structure, propulsion, and tone to the urethra. It comprises two muscle layers: an inner thick longitudinal smooth muscle layer and an outer thin circular smooth muscle layer [257]. 

The pathologies affecting the male urethra may be either congenital, such as hypospadias and epispadias, or acquired, such as strictures. Hypospadias is characterized by the presence of a urethral opening proximal to the glans penis and occurs in 1/300 live male births worldwide [258]. Epispadias is characterized by failure of the urethral plate to tubularize on the dorsum, with the defect ranging from a glandular to a penopubic location. It occurs in 1/117,000 male and 1/150,000 to 1/300,000 female patients [259]. A urethral stricture is a reduction in the urethral caliber resulting from the contraction of scar tissues. It may be due to infection, inflammation, or instrumentation; however, most strictures are idiopathic. About 5 million office visits annually are reported in the US [260]. 

### 4.4. Reconstruction of the Bladder, Ureters and Urethra

Tissue engineering for the reconstruction of the bladder, ureters, and urethra is applied to repair or replace these organs. Reconstructed tissues must be characterized by biocompatibility, functionality, and mechanical properties. They must also support a vascular network and allow intercellular communications, cell adhesion, differentiation, and migration. Scaffold formation is based on biomaterials and cells, mainly stem cells. Full review articles have been recently published and will interest readers [23,53,73,261]. 

#### 4.4.1. Biomaterials

Scaffolds must be biodegradable to allow cell growth and full tissue development and strong enough to support mechanical forces generated by bladder activity and those induced by neighbouring structures. They are formed using synthetic, natural, or hybrid biomaterials or decellularized matrices, seeded or not with cells [23]. 

Synthetic biomaterials such as polyglycolic acid (PGA), polylactic acid (PLA), copoly (lactic/glycolic) acid (PLGA), and poly (ε-caprolactone) (PCL) are mainly used due to their biomechanical properties including biodegradability and biocompatibility. Their main advantages are the fast and reproducible results, the low risk of contamination, the availability, and the low cost. On the other hand, the main disadvantage is that the degradation products of synthetic biomaterials may cause inflammatory reactions, which make the environment inadequate for epithelial cell differentiation, leading to a non-functioning tissue through which urine can pass [53]. Guiding cell response on synthetic biomaterials is a challenging approach that requires the consideration of the substrate’s mechanical properties, biomaterial surface topography, and cell receptor recognition. Biomaterial functionalization is a promising methodology that has been extensively developed. The biomaterial surface is modified with active molecules or substances such as collagen to replicate the native cellular microenvironment and to mimic native ECM-cell interactions. This helps supply cells with adequate biological signals and elicit controlled cell response [262]. 

Natural biomaterials such as collagen type I and silk fibroin exhibit good biomechanical proprieties: elasticity, biodegradability, and biocompatibility. It is possible to combine collagen type I with synthetic materials for better results, creating a hybrid biomaterial characterized by artificial mechanical resistance and natural biocompatibility. On the other hand, decellularized matrices, such as bladder acellular matrix (BAM) and small intestinal submucosa (SIS), are animal-derived matrices or cadaveric organs that are decellularized through chemical, physical, or enzymatic treatments. They allow rapid neovascularization and degrade slowly after implantation. With the refinement of the decellularization techniques, the organization of the ECM is preserved, which is essential to maintain adequate signalling for the cells. For better results, autologous cells must be seeded on the scaffold before implantation. Bone marrow mesenchymal stem cells (BM-MSCs) and urine-derived stem cells (USCs) are alternative cell sources for patients with cancer or benign end-stage bladder diseases. USCs could be seeded on a scaffold and differentiated into urothelial and SMCs with contractile functions, leading to the formation of a multilayered urothelium [23]. 

#### 4.4.2. Cells Used for Urologic Tissue Engineering

Different cell types may be used for ureters and urethra reconstruction. Adult cells, as well as stem cells, are the most used cells. In the rebuilding of the Detrusor muscle layer: adult cells are mainly applied, especially smooth muscle cells from ureter, bladder, or corpus cavernosum biopsies. Smooth muscle-differentiated stem cells such as urine-derived stem cells (USC), adipose-derived stem cells (ASC), and mesenchymal stem cells (MSC) may also be applied. For the reconstruction of the Lamina propria layer: adult cells, such as dermal fibroblasts and oral mucosa fibroblasts, can be used. In addition, stem cells such as bladder mesenchymal cells (BMC) or ASC could be used due to their differentiation potential into secretory-type fibroblasts. For the reconstruction of the urothelium layer: adult cells, such as urothelial cells, oral mucosa keratinocytes, or epidermis keratinocytes, could be adapted. Stem cells could also be adapted, such as ASC [53]. 

## 5. Vagina

### 5.1. Anatomy

Extending from the cervix to the vulva, the vagina is a tubular organ. It has a depth of about 7 to 15 cm. This organ is elastic and muscular. During a woman’s life, the vagina plays an essential role in sexual intercourse, the evacuation of menstrual blood, and childbirth. The microbiome transfer to newborns during birth occurs through the vagina and is hindered by cesarian section delivery [263]. The vagina develops towards the end of the first trimester from the fused Müllerian ducts and the paired sinovaginal bulbs, forming a lumen [264]. The vaginal wall consists of three layers: the inner surface mucosa, the muscularis propria, and the adventitia (Figure 3A). The stratified squamous epithelium rests on the lamina propria. This epithelium fills with intracytoplasmic glycogen in the superficial layers in the presence of estrogen [264], thickening during hormonal peaks during menstruation. Without placental estrogen after birth, the postnatal epithelium atrophies shortly and remains thin until puberty. After menopause, the vaginal epithelium atrophies due to the decline in estrogen levels and keratinization of the surface of the vaginal epithelium results in a similar phenotype to the epidermis [265]. The outer fibres of the vaginal muscle layer extend from the uterus and are longitudinally aligned with this organ. In contrast, the inner fibres form a strong structure aligned in a spiral formation [264]. An acidic microenvironment is created resulting from the colonization of the vagina by lactobacilli, which produce lactic acid from vaginal glycogen, protecting potentially pathogenic bacteria and viruses, such as human immunodeficiency virus (HIV) [266,267]. 

### 5.2. Pathologies

The vagina can be affected by various congenital or acquired defects. Among other malformations, failed or incorrect midline fusion of pelvic structures (bladder, genitalia, colon) can lead to bladder and cloacal exstrophy [268]. In addition, surgical reconstruction using external tissue is often required to correct intersex disorders such as cloacal abnormalities and adrenal hyperplasia [269]. 

MRKH is the agenesis of Müllerian structures. Not only causing agenesis of the uterus, but MRKH women also lack the superior two-thirds of their vaginas [270,271]. As the sinovaginal bulbs cannot fuse to Müllerian structures, there is improper canalization, meaning patients have shallow vaginas, often less than 3.5 cm in depth, with no depth at all in some cases [272]. External structures of the vagina appear normal, meaning diagnosis is usually delayed until puberty, when patients realize a lack of menstrual blood (primary amenorrhea). Associated symptoms are often present, such as renal complications [270]. 

A partial or total vaginal resection may be indicated because of proximal cancers such as cancer of the vagina, cervix, uterus, ovary, rectum, or bladder. In up to 88% of patients, vaginal stenosis, fibrous stenosis, and vaginal shortening are caused by pelvic radiation therapy for these cancers [273]. Vaginal atrophy, hypoestrogenic states, inflammatory and autoimmune diseases, and chemical vaginitis can also cause vaginal stenosis [274]. The standard vaginal structure can also be temporarily or permanently altered by vaginal birth. Most women present damaged supporting tissues after childbirth [275]. Stress urinary incontinence (SUI) often results from structural damage, where 65% of women suffer from SUI. This prevalence increases with age and parity [276]. Women presenting vaginal defects, such as MRKH, suffer from significant psychosexual distress and have been shown to improve with treatment [277,278]. 

### 5.3. Current Surgical Treatment

Although vaginal dilation is the first line of treatment for MRKH women to preserve the native mucosal lining of the vagina (Figure 3B, top), this is a demanding and time-consuming procedure for the patient. It can be traumatizing, especially for young women and must be accompanied by psychological support [279]. Surgery may be an option for vaginal reconstruction in patients unable to undergo vaginal dilation for various reasons or those having failed an attempt at dilation. In 1898, Abbè’s vaginoplasty was introduced for vaginal reconstruction using an autologous skin graft from the inner thighs [280]. The McIndoe method uses a stent to mould the skin graft before implantation and currently is the most prevalent surgical alternative to vaginal dilation. Various approaches use autografted tissues such as bowel sections, oral mucosa, or vulvar flaps for reconstruction (Figure 3B, bottom) [279]. As these techniques avoid dilation, surgeries are possible in a pediatric setting. Inherent problems are associated with heterotopic autografted tissues due to differences in tissue phenotypes. Skin grafts often present vaginal dryness and hair growth, leading to painful intercourse. Bowel segment grafting often causes odor problems due to excessive mucus excretion.

### 5.4. Tissue Engineering

Reconstruction of vaginal tissue with autologous cells has been assayed using several TE methods. Some trials are conducted in a clinical setting [281] (Figure 4). The main advantage of reconstructing vaginal tissue using TE contrary to current procedures is that created implants can be tissue-specific and retain the properties of a native vagina, such as mucus production and adequate microbiome. Main obstacles to patient adherence to currently available treatments such as long-term trauma due to dilation or problems with heterotopic autografts can be avoided using TE. The balance between the pros and cons of treatments should tip in favour of reconstruction using TE. Vaginal construction/reconstruction is usually not an urgent treatment, and the total time required to engineer vaginal substitutes is less important than the potential benefits. Indeed, vaginal dilation is a long process and is not necessarily faster than vaginal TE. Moreover, TE could also serve as a solution for people affected by intersex disorders or trans women. Readers’ interest should be attracted by recently published reviews [65,282]. 

#### 5.4.1. Decellularized Tissues

As previously mentioned, decellularized tissues are promising for providing sufficient mechanical properties if the ECM can be adequately preserved during decellularization. The first attempt at implanting a vaginal substitute used decellularized vaginal or bladder tissue without prior in vitro recellularization, taking advantage of these tissues’ luminal nature [283]. Although re-epithelialization was observed, the graft collapsed shortly after implantation. This study was published in 2002, and decellularization techniques have since been significantly improved; however, preserving the structural integrity of acellular tissues remains challenging. It is known in the literature that acellular implants have a much higher risk of failure than cell-laden tissue-engineered constructs [284]. All subsequent TE vaginal substitutes have been cell-seeded in vitro before implantation.

Raya-Rivera published a clinical trial in 2014, implanting their constructs in four women [285]. Autologous vaginal epithelial and smooth muscle cells were seeded on opposite surfaces on the small intestine submucosa (SIS) to create the graft material. The luminal form was achieved thanks to the natural structure of SIS. Follow-ups have been performed up to 8 years after implantation and have shown promise as a functional TE vaginal substitute, with native-like histology and reported patient satisfaction. Unfortunately, despite apparently good results, the trial was not continued with additional patients, and, to our knowledge, no subsequent clinical trials have taken place.

#### 5.4.2. Synthetic Biomaterials

Synthetic biomaterials, as previously discussed, present the advantage of taking any form given. In vivo implantation studies led by De Filippo, from the same team as Raya-Rivera, have been reported before the clinical trial: a subcutaneous mouse model [286] and an in situ study in rabbits [287]. The native histology was recreated by seeding vaginal epithelial and smooth muscle cells on opposing surfaces of a tubular moulded PGA/PLGA scaffold and then maturing the construct in a bioreactor before implantation. Up to 6 months after implantation, the TE vagina was fully functional with native-like histology. However, the synthetic scaffold had utterly degraded. This team published no follow-up after the 2008 rabbit study [287], as they shifted focus to SIS as a scaffold for clinical evaluation.

#### 5.4.3. Self-Assembly

Using scaffolds in vaginal TE seems promising, with decellularized tissues and synthetic hydrogels proven in in vivo models as potentially viable reconstructive methods. However, most of these models were built more than ten years ago, with no progress since the clinical trial, which had only four participants. Recently, Orabi [288] and Jakubowska [228] proposed a new protocol to engineer vaginal substitutes. The reconstructed tissues were subcutaneously implanted in murine models. This new protocol is derived from the self-assembly protocol previously mentioned. This protocol allows the production of tissues possessing many histological and molecular characteristics of the native vaginal mucosa, notably the presence of a layer of glycogen-storing cells [288]. Moreover, the engineered vaginal mucosae are hormone-responsive, increasing their epithelial thickness in the presence of estradiol, and have served as a model for HIV infection [61]. The study by Jakubowska is fascinating due to the pre-vascularization of the vaginal substitutes and their prolonging in vivo observation to 3 weeks, proving the presence of natural lubrification and several vaginal mucosa-specific markers. Hollow implants were achieved by forming a lumen with two substitutes stitched around an agarose stent. Grafts did not fuse and retained an open lumen after stent removal.

#### 5.4.4. Use of Stents

The previously discussed models would not have been possible without the prolonged use of stents post-operatively to direct vaginal tissue remodelling. Jakubowska reported significant contraction of their grafts as her agarose-based stent dried out, effectively reducing the duration of stent usage [228]. A stent is left continuously for the first three months after surgery in a traditional surgical reconstruction of the vagina using autologous skin grafts. Stent use is progressively reduced to 1 h per day six months after surgery. After that, stent use is still prescribed if regular intercourse is impossible [289]. This prolonged stent use is prescribed due to many years of experience with the McIndoe technique. It should be borrowed to implant any neo-vaginal graft to preserve the desired structure.

## 6. Conclusions

Tissue engineering is a technique that will revolutionize medicine in the 21st century. Much progress has been made in the last thirty years, but much remains to be done. In particular, it will become crucial that discoveries made in laboratories cross the barrier of clinical translation at a higher rate, despite the constant enhancement of regulatory requirements. More and more often, it will become necessary not only to reduce the costs associated with the production of tissues [290] but to rationalize the approaches and to gradually free oneself from the animal products still used for the production of the tissues [52]. Furthermore, until patients can benefit from the most recent advances in tissue engineering and until the shortage of organs to be transplanted has resolved, models produced by tissue engineering should become used as study models for various pathologies. We could replace the old, less sophisticated models: if they have rendered exemplary services, they are gradually becoming obsolete [60]. 

## Figures and Tables

**Figure 1 ijms-24-00009-f001:**
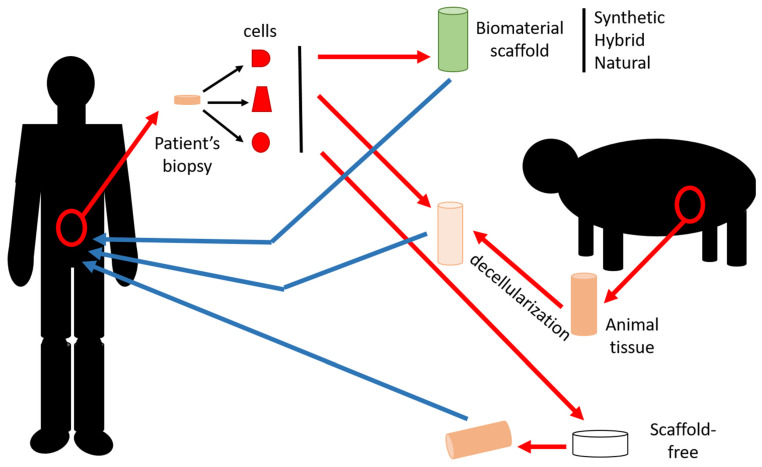
Strategies to engineer tissues. Cells are usually harvested from the target organ to repair/replace or from an organ playing a similar role in the patient. Cell populations can be isolated after extraction or sometimes used mixed. These cells can be seeded onto synthetic, hybrid or natural biomaterial scaffolds produced by different techniques. Alternatively, cells can be seeded onto acellular matrices used as a scaffold. If no scaffold is used, cells can be seeded onto cell culture support and grown in the presence of ascorbate to produce their extracellular matrix. Once the tissue is cell-seeded and matured in a bioreactor, the resulting tissue can be implanted in the patient to repair/replace their failing organ.

**Figure 2 ijms-24-00009-f002:**
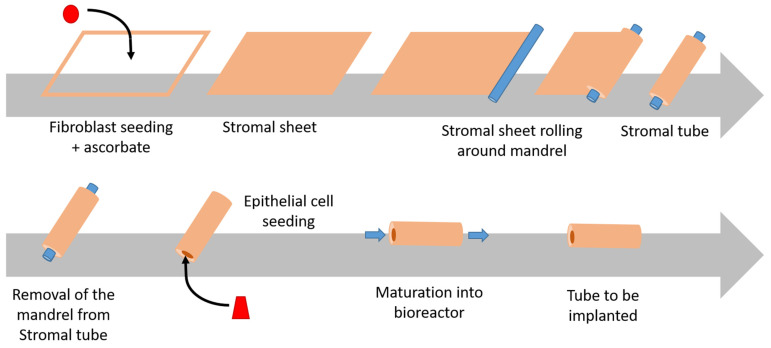
Schema of the production of tubular structure using the self-assembly method. The protocol is described above. It should be noted that ascorbate is maintained during the whole process to allow ECM formation to form the stromal sheet, and also to compensate for ECM degradation due to matrix metalloproteinases (MMP) produced by epithelial cells. The production takes 2 to 3 months to be complete depending on the tissue.

**Figure 3 ijms-24-00009-f003:**
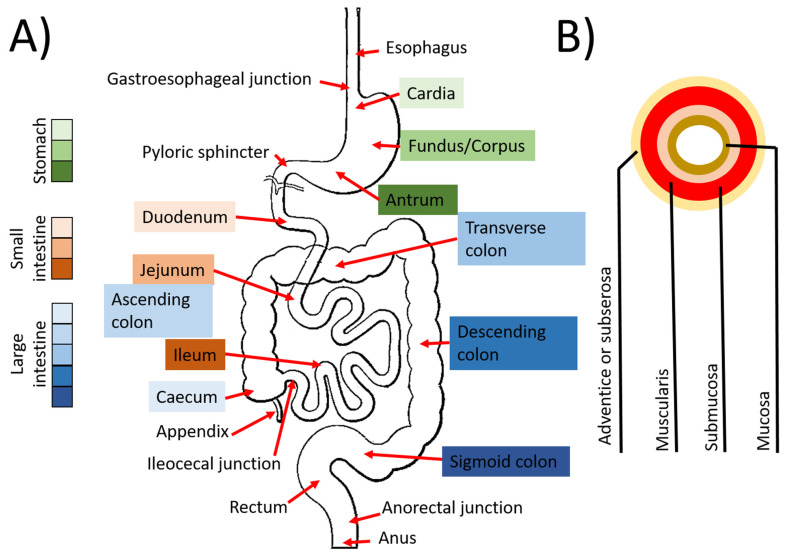
Schema of the digestive tract. (**A**) Gross anatomy of the digestive tract with the different regions. (**B**) Schema of the organization of the tissue. Some regions can vary in width or organization, but the general schema remains roughly the same.

**Figure 4 ijms-24-00009-f004:**
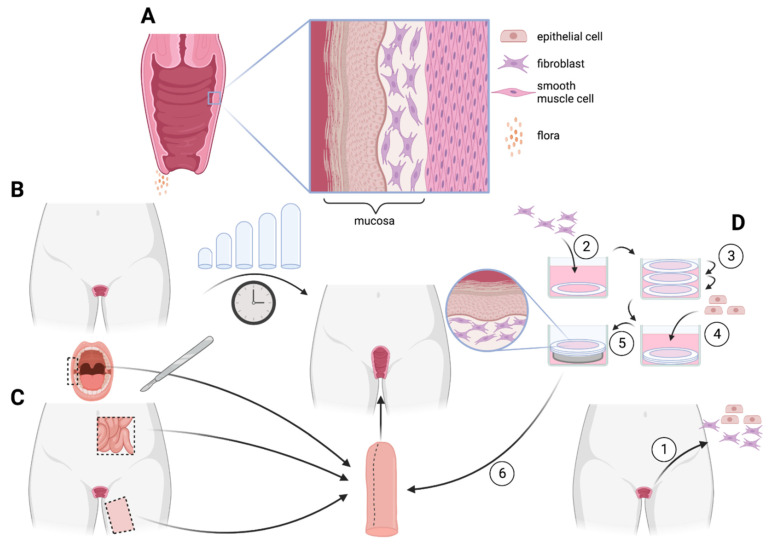
Neovagina construction by techniques: (**A**) Histoanatomy of the vagina. The glycogenated stratified squamous epithelium lines the vaginal lumen, sitting on a stroma. These two layers form the vaginal mucosa, which is followed by the muscularis with shared origin as the uterus. (**B**) Vaginal dilation, Frank’s method, is the gold standard of treatment for MRKH patients. (**C**) Surgical treatments rely on heterotopic autografts. (**D**) Tissue engineering may allow for autologous vaginal mucosa formation in MRKH patients. After vaginal cell extraction (1), the “self-assembly” method is relying on fibroblasts which secrete ECM components when treated with ascorbic acid (2). ECM sheets are piled, fused, (3) and seeded with epithelial cells for native-like autologous vaginal tissue in vitro (4). The constructs are placed at the air/liquid interface to allow epithelial differentation (5) and once matured, it can be grafted into patient (6). This figure was created with BioRender.com and modified from Brownell et al. [282].

**Table 1 ijms-24-00009-t001:** Scaffold-free and synthetic biomaterial protocols for esophageal reconstruction.

Date	1st Author	Biomaterial	Cells	Animal	Ref.
Scaffold-free					
2005	Badylak	n/a	muscle tissue	dog	[102]
2006	Ohki	n/a	oral epithelial cells	dog	[103]
2007	Sakurai	n/a	buccal K	pig	[104]
2011	Honda	n/a	ASC	dog	[105]
2012	Ohki	n/a	oral epithelial cells	human	[106]
2016	Perrod	n/a	ASC cell sheet	pig	[107]
2019	Takeoka	n/a	organoids	rat	[108]
Synthetic biomaterials					
1964	Mark	Teflon	n/a	dog	[109]
2004	Lynen Jansen	PVDF	n/a	rabbit	[110]
2004	Lynen Jansen	Vycril	n/a	rabbit	[110]
2005	Beckstead	PLLA	Ks	in vitro	[111]
2005	Beckstead	PLGA	Ks	in vitro	[111]
2005	Beckstead	PCL/PLLA	Ks	in vitro	[111]
2010	Liang	nitinol	n/a	pig	[112]
2013	Gong	PU	SMCs	rabbit	[113]
2013	Aikawa	BAP	n/a	pig	[114]
2015	Diemer	PCL	SMCs + Epithelial cells	rabbit	[115]
2016	Tan	PLA/PCL	Fbs	in vitro	[116]
2016	Hou	PU	n/a	in vitro	[117]
2016	Park	PCL	MSCs	rabbit	[118]
2016	Tan	PLA/PCL	various	in vitro	[116]
2017	Kuppan	PHBV	ECs + MSCs	in vitro	[119]
2017	Kuppan	PCL	ECs + MSCs	in vitro	[119]
2017	Dorati	PLA/PCL	Fbs	in vitro	[120]
2017	Tan	PLGA	various	in vitro	[121]
2018	La Francesca	PU	ASCs	pig	[122]
2018	Wei	PU-derived	MSCs	in vitro	[123]
2018	Chung	PCL	Fbs	rat	[124]
2019	Jensen	PU	pig Epithelial cells + MSCs	pig	[125]
2019	Kim	PU/PCL	MSCs	rat	[126]
2019	Zhuravleva	Polyamide 6	MSCs, ASCs	in vitro	[127]
2019	Soliman	PU	MSCs + SMCs	in vitro	[128]
2019	Kim	PU/PCL	MSCs	rat	[126]
2020	Pisani	PLA/PCL	MSC	in vitro	[129]

ASC = Adipose-derived stem cell; EC = Endothelial cell; Fb = Fibroblast; K = Keratinocyte; MSC = Mesenchymal stem cell; SMC = Smooth muscle cell.

**Table 2 ijms-24-00009-t002:** Hybrid and natural biomaterial protocols for esophageal reconstruction.

Date	1st Author	Biomaterial	Cells	Animal	Ref.
Hybrid biomaterials					
1991	Purushotham	Vycril/Coll	n/a	pig	[130]
1993	Natsume	Silicone/Coll	n/a	dog	[131]
1995	Takimoto	Silicone/Coll	n/a	dog	[132]
2003	Grikscheit	PGA/collagen	organoids	rat	[133]
2005	Zhu	PLGA/Coll	SMCs	in vitro	[134]
2006	Zhu	PLLC/Coll	various	in vitro	[135]
2007	Zhu	PLLC/Fibronectin	Epithelial cells	in vitro	[136]
2008	Nakase	Amniotic + PGA	oralKs + Fbs	dog	[137]
2014	Lv	PCL/silk	n/a	rabbit	[138]
2017	Kuppan	PHBV/gelatin	ECs + MSCs	in vitro	[119]
2017	Kuppan	PCL/gelatin	ECs + MSCs	in vitro	[119]
2017	Dorati	PLA/PCL + Chitosan	Fbs	in vitro	[120]
2020	Nam	PCL + ECM bioink	SMCs + Epithelial cells	in vitro	[139]
Natural biomaterials					
1998	Takimoto	collagen	n/a	dog	[140]
1999	Yamamoto	collagen	n/a	dog	[141]
1990	Natsume	collagen	Mucosal cells	dog	[142]
2001	Kajitani	Elastin	n/a	pig	[143]
2002	Komuro	collagen	n/a	pig	[144]
2021	Gundogdu	silk	n/a	pig	[145]

EC = Endothelial cell; Fb = Fibroblast; K = Keratinocyte; MSC = Mesenchymal stem cell; SMC = Smooth muscle cell.

**Table 3 ijms-24-00009-t003:** A cellular matrix protocols for esophageal reconstruction.

Date	1st Author	Biomaterial	Cells	Animal	Ref.
Acellular matrices					
2000	Badylak	SIS	n/a	dog	[146]
2000	Badylak	Bladder submucosa	n/a	dog	[146]
2001	Isch	Dermis	n/a	dog	[147]
2005	Badylak	Pig bladder	n/a	dog	[102]
2005	Beckstead	Dermis	Ks	in vitro	[111]
2006	Marzaro	Esophagus	SMCs	pig	[148]
2006	Lopes	SIS	n/a	rat	[149]
2006	Bhrany	Esophagus	Epithelial cells	rat	[150]
2007	Urita	Gastric	n/a	rat	[151]
2008	Bhrany	Esophagus (crosslink)	Epithelial cells	rat	[152]
2009	Nieponice	Pig bladder	n/a	dog	[153]
2009	Wei	SIS	Oral epithelial cells	dog	[154]
2009	Doede	SIS	n/a	pig	[155]
2010	Gaujoux	Aorta	n/a	pig	[156]
2011	Badylak	SIS	n/a	human	[157]
2011	Clough	SIS	n/a	human	[158]
2012	Hoppo	SIS	n/a	human	[159]
2013	Tan	SIS	BMSCs	dog	[160]
2013	Keane	Esophagus	Perivascular SCs	rat	[161]
2013	Totonelli	Pig esophagus	n/a	in vitro	[162]
2014	Nieponice	Bladder submucosa	n/a	human	[163]
2015	Poghosyan	SIS + HAM	Myoblasts + Epithelial cells	pig	[164]
2017	Urbani	Rabbit Esophagus	n/a	in vitro	[165]
2017	Okuyama	Biosheet	n/a	dog	[166]
2017	Catry	SIS	MSCs	pig	[167]
2018	Luc	Esophagus	ASCs	pig	[168]
2018	Urbani	Rat Esophagus	Various	mouse	[169]
2020	Marzaro	Esophagus	MSCs	pig	[170]
2021	Chaitin	Esophagus	ESCs + Fbs	in vitro	[171]

SIS = Small intestinal segment; HAM = Human amniotic membrane; ASC = Adipose-derived stem cell; BMSC = Bone marrow stem cell; EC = Endothelial cell; ESC = Embryonic stem cell; Fb = Fibroblast; K = Keratinocyte; MSC = mesenchymal stem cell; SC = stem cell; SMC = Smooth muscle cell.

**Table 4 ijms-24-00009-t004:** Protocols for the reconstruction of stomach or stomach patches.

Date	First Author	Biomaterial	Cells	Animal	Ref.
Scaffold free					
2010	Stange	n/a	organoids	in vitro	[179]
2017	Tanaka	n/a	Myoblast sheet	rat	[180]
Synthetic biomaterials					
2009	Sala	PGA/PLLA	Stomach cells	pig	[181]
2012	Maemura	PGA/PLLA	Epi organoids	rat	[182]
2012	Maemura	PGA	organoids	rat	[182]
Hybrid biomaterials					
2001	Hori	PGA/coll	n/a	dog	[183]
2002	Hori	PGA/coll	n/a	dog	[184]
2003	Griksheit	PGA/coll	Ep iorganoids	rat	[185]
2004	Maemura	PGA/PLLA/Coll	Epi organoids	rat	[186]
2008	Maemura	PGA/PLLA/Coll	Epi organoids	rat	[187]
2009	Araki	PLC/PGA/Coll	n/a	dog	[188]
2009	Araki	PDLCL/PGA/Coll	n/a	dog	[188]
2011	Speer	PGA/PLLA/Coll	organoids	mouse	[189]
Natural biomaterials					
2010	Barker	Matrigel	organoids	in vitro	[190]
2013	Katano	collagen	Epi	in vitro	[191]
2014	McCracken	Matrigel	Human iPSC	in vitro	[192]
2015	Bartfeld	Matrigel	organoids	in vitro	[193]
2015	Schumacher	Matrigel	organoids	in vitro	[194]
2015	Noguchi	Matrigel	mouse ESC	in vitro	[195]
2016	Schlaermann	Matrigel	organoids	in vitro	[196]
2017	McCracken	Matrigel	iPSC	in vitro	[197]
Acellular matrices					
2007	Ueno	SIS	n/a	rats	[198]
2015	Nakatsu	SIS	MSC	rat	[199]

Epi = epithelial cell; ESC = Embryonic stem cell; iPSC = Induced pluripotent stem cell; MSC = mesenchymal stem cell.

**Table 5 ijms-24-00009-t005:** Protocols for intestines reconstruction.

Date	1st Author	Biomaterial	Cells	Animal	Organ	Ref.
Scaffold-free						
2012	Yui	n/a	organoids	mouse	Int	[207]
2014	Zani	n/a	AFSC	rat	Int	[208]
2021	Sugimoto	n/a	organoids	rat	Int	[209]
Synthetic biomaterials						
2015	Finkbeiner	PGA/PLLA	organoids	mouse	Sint	[210]
2017	Schlieve	PGA	iPSC+organoids+neurons	mouse	Sint	[204]
2014	Costello	PLGA	Cell lines/organoids	in vitro	SInt	[211]
1997	Choi	PGA	organoids	mouse	int	[212]
2004	Grikscheit	PGA	organoids	rat	Int	[213]
2009	Sala	PGA/PLLA	Int cells	pig	Int	[181]
2015	Liu	PGA	organoids	mouse	Int	[214]
2016	Gjorevski	PEG-VS	murine crypt cells	in vitro	Int	[215]
2019	Liu	PGA	crypt SC	rat	Int	[216]
2019	Liu	PCL	crypt SC	rat	Int	[216]
Hybrid biomaterials						
2013	Levin	PGA/collagen	organoids	mouse	Sint	[217]
2016	Wieck	PGA/PLLA/collagen	organoids	mouse	Int	[218]
2019	Ladd	PGS/collagen	organoids	mouse	Int	[219]
Natural biomaterials						
2014	Watson	collagen	iPSC/ESC organoids	mouse	Sint	[220]
2014	Fukuda	Matrigel	organoids	mouse	Sint	[221]
2016	Cromeens	Matrigel	murine crypt cells	in vitro	SInt	[222]
2017	Workman	collagen	iPSC/ESC organoids	mouse	Sint	[205]
2017	Zakhem	Chitosan	human cells	rat	Int	[223]
2019	Liu	Collagen	crypt SC	rat	Int	[216]
2018	Sugimoto	Matrigel	human organoids	mouse	Int	[224]
2019	Capeling	Alginate	human HPSC	in vitro	Int	[225]
Acellular matrices						
2012	Totonelli	Rat SInt	AFSC	in vitro	Sint	[226]
2015	Finkbeiner	Pig Int	organoids	mouse	SInt	[210]
2017	Kitano	Rat Int	iPSC	rat	Int	[227]
2020	Palikuqi	Rat Int	various	mouse	Int	[228]
2020	Meran	Int	Fb + organoids	mouse	Int	[229]

AFSC = Amniotic fluid stem cell; ESC = Embryonic stem cell; Fb = Fibroblast; iPSC = Induced pluripotent stem cell; HPSC = Human pluripotent stem cell; SC = stem cell; Sint = small intestine; Int = Intestine.

## Data Availability

Not applicable.

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
