# Peer review of "Tissue Engineering for Gastrointestinal and Genitourinary Tracts"

_ijms, 2022, doi:10.3390/ijms24010009_

Round 1
Reviewer 1 Report
I read with interest this extensive review about tissue engineering of gastrointestinal and genitourinary tract.
This manuscript has many strong points.
1. It is written in a understandable and plain English language. The authors are not native speakers ( Canadians from French speaking region) however the language is quite in a high level and gives the readers a pleasure to read this interesting article.
2.It is very analytical and to the point reports the various strategies and techniques in different organs.
3. The figures are offering a good description and exhibition of the mechanisms of tissue engineering.
4.It offers information in readers who are not familiar.
My only remark is the in the tables, the word auteur should be replaced with the word author.
I congratulate the authors for their hard work.
Reviewer 2 Report
Dear Authors,
it was a pleasure to read your manuscript. Your deep insight into the field is very comprehensive. Nevertheless, I think you miss the important role of Wharton Jelly derived MSCs which has tremendous potential, are cheap and easy to collect and are ethically clean comparing to embryonic cells.
What is even more important, I think you miss a paragraph concerning conditions of MSC's culture (physiologic hypoxia, etc.) - establishing worldwide protocols for that is essential for the proper comparison of results in this field.
Please address those two queries.
